# StagFormer: Time Staggering Transformer Decoding for Running Layers In Parallel

## Abstract

Standard decoding in a Transformer based language model is inherently sequential as we wait for a token's embedding to pass through all the layers in the network before starting the generation of the next token. In this work, we propose a new architecture StagFormer (Staggered Transformer), which staggered execution along the time axis and thereby enables parallelizing the decoding process along the depth of the model. We achieve this by breaking the dependency of the token representation at time step $i$ in layer $l$ upon the representations of tokens until time step $i$ from layer $l-1$. Instead, we stagger the execution and only allow a dependency on token representations until time step $i-1$. The later sections of the Transformer still get access to the "rich" representations from the prior section but only from those token positions which are one time step behind. StagFormer allows for different sections of the model to be executed in parallel yielding up to 33% speedup in decoding while being quality neutral. We also explore many natural variants of this idea. We present how weight-sharing across the different sections being staggered can be more practical in settings with limited memory. We show how one can approximate a recurrent model during inference using such weight-sharing. We explore the efficacy of using a bounded window attention to pass information from one section to another which helps drive further latency gains for some applications. We also explore demonstrate the scalability of the staggering idea over more than 2 sections of the Transformer.

## 1 Introduction

Transformers (Vaswani et al., 2017) have seen tremendous success as the primary backbone for language models (Chowdhery et al., 2022; Hoffmann et al., 2022; Brown et al., 2020). The architecture lends itself particularly well for causal language modeling by allowing efficient, highly parallelized training over large datasets. Moreover, the model can be efficiently partitioned across multiple devices (Pope et al., 2022) enabling model parallelism across machines. However, it is well known that, during inference, decoding from a Transformer is an inherently sequential task. This task becomes more expensive when trying to decode long sequences due to the cost of attention, which scales linearly with respect to sequence length.

There have been numerous works which try to make inference more efficient in practice. Speculative decoding, local attention and other efficient attention variants (Tay et al., 2022), KV cache optimizations, blockwise parallel decoding (Stern et al., 2018) etc. are a few such works. However, there haven't been many works which try to tackle the sequentiality imposed by the depth of the Transformer. Depth, while known to be essential for the strong performance of Transformers (Raffel et al., 2023; Zhao et al., 2023; Ye et al., 2024), introduces a proportional cost in terms of decoding latency.

In this work, we take a look at how we can introduce some level of parallel execution along the depth axis of a Transformer language model while decoding.

We introduce StagFormer (*Staggered Transformer*), a novel Transformer variant which breaks the sequential dependency of the upper layers on the lower layers by *staggering* the time dependency of token embeddings passed from the lower layers to the upper layers. In particular, we present a mechanism by which, at time step $i$, the upper layers of the model use the rich representations of tokens computed by earlier layers only until time step $i-1$. Note that in a standard Transformer this

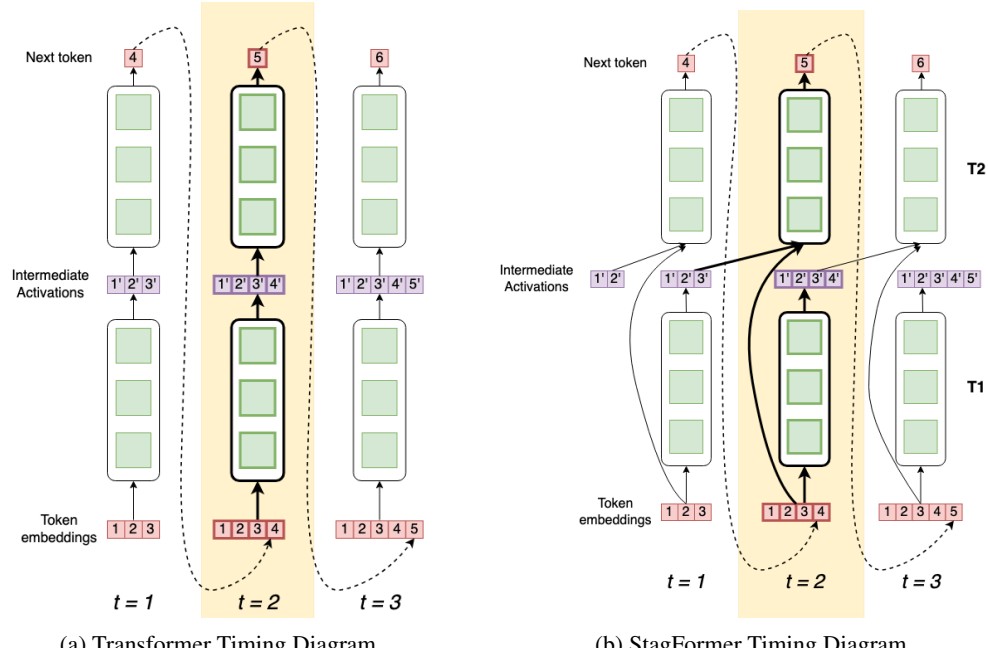

(a) Transformer Timing Diagram   (b) StagFormer Timing Diagram

Figure 1: Depiction of forward pass in a standard Transformer compared with that of StagFormer. Note that in StagFormer, the data dependency in a given time step has been broken for the two stacks T1 and T2.

dependency is allowed until time step $i$. StagFormer is a deviation from the traditional Transformer and as such requires to be trained from scratch to respect the staggering during decoding. We show how one can train and decode efficiently using our architecture.

We perform language modeling experiments on the Pile dataset (Gao et al., 2020) with StagFormer and show that we can get significant latency savings during decode due to parallel execution of different parts of the Transformer stack without taking a hit in quality. Finally, we also explore many useful variants of the StagFormer architecture and demonstrate their efficacy for language modeling. We include a thorough downstream task evaluation for our trained language models across a suite of tasks.

Table 1: StagFormer vs Standard Transformer: Pretrained on the Pile dataset for 300B tokens.

| Model | Pile Pplx. | HellaSwag | ARC-E | ARC-C | WinoGrande | SuperGLUE | MBPP | Lambada | SQuADv2 | GEM-XSum rouge2 | Avg. |
|---|---|---|---|---|---|---|---|---|---|---|---|
| Baseline (18L) 1.6B params | 4.026 | 49.8 | 60.1 | 31.8 | 53.4 | 59.3 | 0 | 3.7 | 31.8 | 0.9 | 32.3 |
| Baseline (36L) 2.8B params | 3.780 | 53.3 | 66.7 | 34.6 | 60.4 | 62.1 | 0.2 | 10.5 | 36.3 | 1.6 | 36.2 |
| StagFormer $p = 2$ Separate-Weights (2 x 18L Stacks) 2.9B params | **3.756** | **58** | **66.8** | **36.3** | **60.5** | 61.3 | **1.6** | **18.5** | **44.4** | 1.5 | **38.8** |

## 1.1 RELATED WORK

The Transformer was originally proposed in the seminal work of Vaswani et al. (2017). Decoder-only language modeling using the Transformer was originally proposed by Radford (2018) and has since become a standard backbone to many frontier language models today.

There has been an enormous body of research dedicated towards making Transformer training or inference more efficient (Tay et al., 2022). These involve approaches which focus on pre-training such as distillation(Xu et al., 2024), layer stacking (Panigrahi et al., 2024), Alternating-updates (Baykal et al., 2024), Matryoshka Transformer (Kusupati et al., 2022) among others. Quantization (Xiao

et al., 2023) has been another widely successful approach at speeding up inference of language models. There have been other approaches specifically focused on improving the decoding speed from language models such as speculative decoding and related works (Leviathan et al., 2023; Sun et al., 2024; Santilli et al., 2023).

There has also been a huge body of work focusing on making the self-attention more efficient. Some of these works have introduced the idea of introducing a form of recurrence mechanism into models, such as Transformer-XL and State Space Models (SSMs) like Mamba (Dai et al., 2019; Gu et al., 2022; Gu & Dao, 2024). Block-Recurrent Transformers use cross-attention to introduce a per-layer recurrence mechanism into Transformer networks (Hutchins et al., 2022).

More closely related to our effort are works such as Medusa (Cai et al., 2024) which uses parallel heads to decode multiple tokens ahead at once, Staircase Attention (Ju et al., 2022) which uses a similar idea of staggering attention window context as we advance deeper into the Transformer stack. However, they mainly explore a variant of the idea which allows one to bring in the benefits of RNNs rather than efficiency gains, our main focus here.

Our shared-weight variant of StagFormer is closely related to the idea of a *looped* Transformer, where the hidden activation signals are sent through the layers of the network multiple times (Dehghani et al., 2018; Giannou et al., 2023; Gatmiry et al., 2024). Part of the intuition behind looping is that the lower layers of a network can reuse the more-information-rich activations from layers later in the same network in the next iteration of the loop to create higher quality representations. A key difference in our method from looping is that it breaks the strict data-dependency on each prior loop, allowing for parallel execution of different passes through the network.

## 2 STAGGERED TRANSFORMERS (STAGFORMER)

In this section we describe our Staggered Transformer (StagFormer) architecture. We begin with a brief background on a decoder-only language models based on the standard Transformer, the backbone for most state-of-the-art language models today.

**Language Modeling with the Transformer**  A Transformer of depth $\ell$ is a sequence-to-sequence model which takes in a token sequence of length $N$ and generates an output sequence of length $N$. The tokens are each first mapped to a $d$-dimensional representation using an embedding layer. Positional information may also be combined into the embedding at this stage. Denote the token embeddings so obtained by $\mathbf{t}_0^{1,\ldots,N}$. Then, these representations are progressively modified by applying a sequence of Transformer layers, $L_1, \ldots, L_\ell \colon \mathbb{R}^d \to \mathbb{R}^d$ iteratively: $\mathbf{t}_j^{1,\ldots,N} = L_j\left(\mathbf{t}_{j-1}^{1,\ldots,N}\right)$ for $j \in \{1, \ldots, \ell\}$. Each layer $L_j$ consists of two main operations: (a) self-attention which combines information across the different token embeddings and (b) a feed-forward network which modifies each individual token embedding. The two main operations are applied along with residual connections and layer normalization. There may additionally be a position encoding incorporated into the embedding during self-attention stage as well.

To use Transformers as decoder-only language models, a popular paradigm is that of causal language modeling. Given a train dataset of examples each of which is a sequence of tokens of length $N$, causal language modeling simultaneously trains to minimize $N$ loss terms on each sequence. These loss terms minimize the cross-entropy loss of the model's prediction for token $\mathbf{t}^i$ using the prefix $\mathbf{t}^{1,\ldots,i-1}$. During training, all $N$ of these loss terms can be evaluated in parallel with the use of causal masking. During decoding, the model iteratively generates one new token at a time by passing token $\mathbf{t}_i$ through the $\ell$ layers sequentially to obtain $\mathbf{t}_{i+1}$. This means that growing the network depth incurs a linear cost on the time taken to decode the next token during inference. However, there is ample evidence that depth is crucial for good quality models (Devlin et al., 2019; Raffel et al., 2023). There is fundamentally no way to avoid this cost in a Transformer, since every token relies on the completed predictions of every other prior token.

**StagFormer**  StagFormer introduces a way to break the sequential dependency of layers within a Transformer network and still be able to perform efficient and performant causal language modeling. We first partition our $\ell$ layers into $p$ sub-networks we call *stacks*. For ease of exposition we will first focus on the simplest case $p = 2$. Let $h = \lfloor \ell/2 \rfloor$. StagFormer allows for execution of the stacks

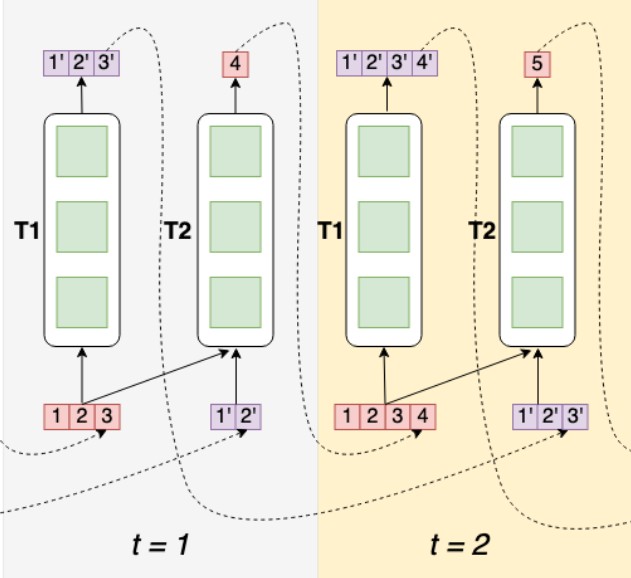

Figure 2: Depiction of the parallel execution of stacks T1 and T2 in a 2-stack StagFormer. In a given time step, both T1 and T2 can run in parallel: T1 producing the intermediate activation to be used in the next time step and T2 producing the output token for the next time step.

of layers $1, ...h$ and $h + 1, ..., \ell$ in parallel in a given time step $i$ by *staggering* the dependency between $\mathbf{t}_h^i$ and $\mathbf{t}_{h+1}^i$. In particular, we compute $\mathbf{t}_{h+1}^i$ as a function of the original token sequence $\mathbf{t}_0^{1,...,i}$ and the $h^{th}$ layer representations taken until time step $i - 1$: $\mathbf{t}_h^{1,...,i-1}$. Crucially we exclude a dependency on $\mathbf{t}_h^i$. This allows the lower half of layers to begin computing the predictions for the next token in the sequence, $\mathbf{t}_h^{i+1}$, while the upper layers in the network are finishing computing the final the prediction for position $i$, $\mathbf{t}_\ell^i$.

We realize this by passing the original token embedding, $\mathbf{t}_0^i$ as input to the second half of the layers, $L_{h+1}, \ldots, L_\ell$, and by augmenting these layers with cross attention to the final activations of the first half of the network on the prior tokens, $\mathbf{t}_h^1, \ldots, \mathbf{t}_h^{i-1}$, when computing the final predictions for the next token after position $i$. Thus $\mathbf{t}_{h+1}^i$ does not actually depend on the prior layers' representation of the token, $\mathbf{t}_h^i$, it is a function of the initial token embedding, $\mathbf{t}_0^i$, and cross-attends to the previous layers' representations of only past tokens, $\mathbf{t}_h^0, \ldots, \mathbf{t}_h^{i-1}$.

Figure 1 shows a timing diagram of how decoding works in StagFormer. The parallel execution of the two stacks is shown more clearly in Figure 2. During training, to faithfully simulate StagFormer's decoding, we sequentially pass our token sequence over the two stacks of layers where we allow the second stack to cross-attend to the outputs of the first stack with masking such that at position $i$ we can only cross-attend to the first $i - 1$ outputs from the first stack. This completes a description of how we can train and decode using StagFormer. The full algorithm is given is Algorithm 1.

This idea can be generalized to $p$ partitions of the $\ell$ layers by having each new partition stagger an additional time-step. We call this technique *staggering* the Transformer network over $p$ stacks. A full description of this generalization is presented in Section 3.4.

The main advantage of StagFormer is the potential to save latency during decoding by executing stacks in parallel. This can be realized efficiently on today's hardware accelerators such as TPUs and GPUs. Staggering the dependency on the processed representations of tokens until time step $i$ between the first and second stacks of StagFormer can, in principle, lead to a decrease in quality of the model. However, the additional cross-attention parameters in the second stack help ameliorate this decline. In Section 4, we train and evaluate StagFormer for language modeling and observe that a depth $\ell$ StagFormer with 2 stacks outperforms a depth $\ell$ regular Transformer (Table 1) while giving a decode latency speedup of $33\%$ as shown in Table 2. Overall, we see strong performance

---

**Algorithm 1** StagFormer algorithm

---

**Input:** $\mathbf{t}_0^1, \ldots, \mathbf{t}_0^i \in \mathbb{R}^d$ : Token embeddings for positions $1, \ldots, i$ in the input sequence.
**Output:** $\mathbf{t}_\ell^i \in \mathbb{R}^d$ : The predicted token embedding for position $i + 1$ in the input sequence where $\ell$ is the total number of Transformer layers in the network.

1: **First pass** : for each layer $L_1, ..., L_h$ where $h \equiv \lfloor \ell/2 \rfloor$ compute $\mathbf{t}_j^i = L_j \left( \mathbf{t}_{j-1}^{1,\ldots,i} \right)$.
   Each application of $L_j$ using standard Transformer layer with self-attention and feed-forward layers.

2: **Second pass** : for each layer $L_{h+1}, \ldots, L_\ell$ compute $\mathbf{t}_j^i = L_j' \left( \mathbf{t}_u^{1,\ldots,i}, \mathbf{t}_h^{1,\ldots,i-1} \right)$.
   Where $u = 0$ when $j = h + 1$ and $u = j$ otherwise.
   Where $L_j'$ is a Transformer layer that has an additional cross-attention layer between the self-attention and feed-forward layers that uses $\mathbf{t}_h^{1,\ldots,i-1}$ for KV inputs.

3: **Return** $\mathbf{t}_\ell^i$

---

gains on tasks such as SQuADv2, Lambada and HellaSwag while being neutral with the baseline on some others such as SuperGLUE.

Table 2: Latency Benchmarking for a baseline Transformer vs a comparable quality StagFormer model. While we suffer a modest increase in prefill latency, the per step decode latency speeds up by **33%** leading to significant savings during decoding. Benchmarking was performed on 16 TPUv5e chips.

| Model | Total prefill time for 1024 tokens (ms) | Average decode time for 1024 tokens (ms) |
|---|---|---|
| Transformer 36L | 5.45 | 2.06 |
| StagFormer 2x18L | 6.66 | **1.55** |

In the next section, we describe some variants of the StagFormer architecture which might be more applicable in certain settings.

## 3 EXTENSIONS OF THE STAGFORMER

In this section, we describe certain natural extensions and variants of the StagFormer architecture.

### 3.1 SHARED-WEIGHTS STAGFORMER

In scenarios where we are bound tightly on memory requirements, one can use a variant where we share weights across the different stacks being staggered. Such weight sharing lowers the quality of the model but can save significantly on memory requirements and can be more applicable in memory-constrained settings. Here we use the same weights for self-attention and feed-forward layers for both the passes. The cross-attention weights are the only unique weights for the second pass. So for some input $\mathbf{t}_0^i$, we would apply $L_1, \ldots, L_\ell$ twice. The first pass processes the input as a standard Transformer network, alternating self-attention and feed-forward layers. The second pass introduces cross-attention layers which allow each token to attend to the final activations of all prior tokens, $t_L^1, \ldots, t_L^{i-1}$.

During inference, we can have the networks execute the two passes in parallel. This is because, like separate-weights StagFormer, the second pass only depends on the final activations of prior tokens and both passes operate on the same input. The results with shared weights StagFormer are presented in Table **??**. We would like to remark that a 2 stack shared-weight StagFormer with each stack having 18 layers performs significantly better than a 18 layer baseline model which has a similar number of parameters. Therefore, StagFormer is an effective way of boosting the performance given a parameter budget.

Note that shared-weights StagFormer is more similar to looped Transformers than the separate-weights variant, but with an additional cross-attention layers acting as a recurrence mechanism. Extending this idea during inference, once the model has finished processing the prefix, we show that we can use cross-attention to the final activations of the prior tokens to approximate recurrent inference requiring only the second pass in section 3.2.

## 3.2 SHARED-WEIGHTS STAGFORMER APPROXIMATES A RECURRENT MODEL

One method we explore for decoding with shared-weights StagFormer is to use the cross-attention to the final activations of prior tokens as a recurrence mechanism. Rather than having the network process each token twice in parallel, with only the second pass using cross-attention, we only have the network operate on each input during decoding once. When doing so, the network cross-attends to the final activations of all prior tokens.

This method of decoding resembles a recurrent neural network (RNN) where the final activations of prior tokens are the RNN's hidden state and cross-attention serves as a gating mechanism while processing the current token.

We show that it is possible to use shared-weights StagFormer for recurrent decoding using this scheme, even when the model is trained using two separate passes. However, we find that the generated text's quality is not as good as when we process decode new tokens the original way, with two networks running in parallel.

---

**Algorithm 2** Recurrent Decoding using Shared-Weights StagFormer

**Input:** $\mathbf{t}_0^1, \ldots, \mathbf{t}_0^i \in \mathbb{R}^d$ : Token embeddings for positions $1, \ldots, i$ in the input sequence.
**Output:** $\mathbf{t}_\ell^i \in \mathbb{R}^d$ : The predicted token embedding for position $i+1$ in the input sequence where $L$ is the total number of Transformer layers in the network.

1: **Prefill** : Use the shared-weights StagFormer algorithm to process the prefix (Algorithm 3).

2: **Decoding** : for each layer $L_1, \ldots, L_\ell$ compute $\mathbf{t}_j^i = L'_j \left( \mathbf{t}_{j-1}^{1,\ldots,i}, \mathbf{t}_l^{1,\ldots,i-1} \right)$.

   Where $L'_j$ has an additional cross-attention layer between the self-attention and feed-forward layers to the Transformer layers in the first pass that uses $\mathbf{t}_l^{1,\ldots,i-1}$ for KV inputs. The rest of the parameters in $L'_j$ are the same as those in $L_j$ used for the prefill.

3: **Return** $\mathbf{t}_\ell^i$

---

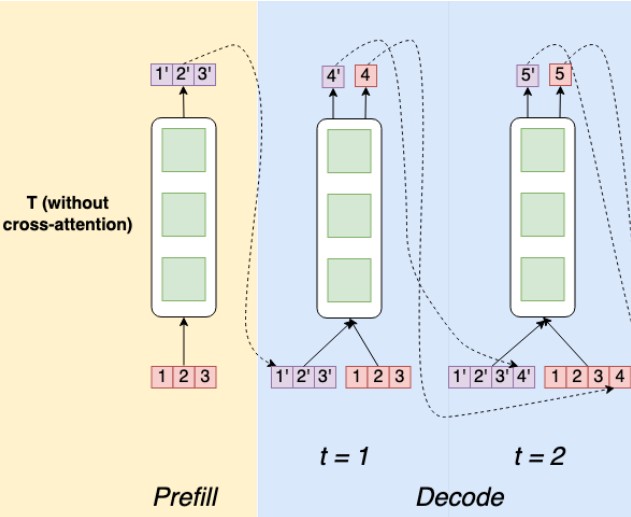

Figure 3: Timing Diagram of Prefill vs Decode steps for Recurrent Inference with Shared-Weights StagFormer. During prefill, the Transformer T is run without cross-attention and during decode it is run with cross-attention.

### 3.3 STAGFORMER WITH LOCAL CROSS-ATTENTION

If we want stronger latency savings and are willing to take a slight quality hit, a further optimization for StagFormer that is simple to implement is to use local attention for the cross-attention between passes (Beltagy et al., 2020). We observe that StagFormer still performs well even when using local cross-attention with relatively small attention window sizes. StagFormer is also capable of giving non-trivial quality when using an attention window size of 1, which converts the application of the cross-attention in layer $L_j$ on token $\mathbf{t}_{j-1}^i$ to a linear function of $\mathbf{t}_h^{i-1}$ (recall $h \equiv \lfloor \ell/2 \rfloor$).

Section 4.3 shows the impact of using local attention with window sizes 512, 128, and 1 on Stag-Former's performance on pretraining perplexity and downstream tasks. We show local attention can be used successfully with both the separate-weights and shared-weights variants.

### 3.4 STAGFORMER WITH MORE THAN TWO STACKS

A natural extension of StagFormer idea we had touched upon earlier is to have $h$ be less than $\lfloor \ell/2 \rfloor$ and to stagger over more than 2 stacks through the network. For instance, we could have $h \equiv \lfloor \ell/3 \rfloor$ and stagger the network over 3 stacks. Let $p$ be the number of stacks we stagger the network over, then $h \equiv \lfloor \ell/p \rfloor$. Intuitively, as we increase the number of stacks $p$, due to progressive staggering, at time step $i$ stack $s$ only gets to see tokens until time step $i - p + s$ but needs to produce activations which help predict token $i + 1$. Thus the job becomes more difficult to learn as $p$ increases, and the depth of each stack reduces which contributes to eventual degradation in quality.

Our experiments indeed find that model quality suffers when $p > 2$. However, we find that we can recover significantly by imploring a simple change for StagFormer when $p > 2$. Rather than using just the final stack's output for computing the final logits, we use a linear combination of each stack's output with learnable coefficients, $\alpha_1, \ldots, \alpha_p$. Algorithm 4 defines separate-weights StagFormer for when $p > 2$ in the Appendix.

Our experiments ablate the linear combination at the end of separate-weights StagFormer when $p > 2$ to demonstrate its effectiveness. Our results are summarized in Section 4.4. We find that as we increase $p$ model quality suffers, but we are able to recover some of the lost performance by using a linear combination of each stack's output. We explored the settings of $p = 3, 4$ here, but there might be ways to extend the approach effectively to even larger values of $p$ which we leave for future work.

**Shared-Weights StagFormer with More Than Two Passes**  One can also increase the number of staggered passes with shared-weights StagFormer. Since the Transformer layer weights are shared between passes, shared-weights StagFormer would process the same input multiple times, cross-attending to prior tokens' final activations from prior passes. We find that doing so increases model quality, even without using the linear combination of outputs that separate-weights StagFormer uses when $p > 2$. Our results are summarized in Table 4.

## 4 EXPERIMENTS

In this section, we describe our pre-training downstream evaluation setup we used for the different variants of the StagFormer via causal language modeling on the Pile dataset (Gao et al., 2020). We begin by outlining our experiment setting. We also demonstrate the performance of various extensions covered in Section 3.

### 4.1 EXPERIMENTAL SETTING

We performed our experiments using a standard Transformer architecture. The model uses a vocabulary size of 256,000. The model adds global positional embeddings to initial token embeddings and applies Rotary Positional Embeddings (RoPE) in the attention layers (Su et al., 2023). We compare StagFormer to an 18 layer baseline model with 1.6 billion parameters as well as a baseline where we double the number of layers, resulting in a 2.8 billion parameter model. We pretrained our model on The Pile dataset with a global batch size of 1024 and a max sequence length of 1280 (Gao et al.,

2020). We trained the model for $250,000$ steps or 327 billion tokens which Gu & Dao (2024) demonstrated should be enough tokens for the model to develop few-shot learning capabilities.

We evaluate the model's performance on several few-shot learning tasks (Brown et al., 2020). The evaluation benchmarks include HellaSwag, ARC-E/C, WinoGrande, SuperGLUE, MBPP, Lambada, SQuADv2, and others (Zellers et al., 2019; Ma et al., 2023; Sakaguchi et al., 2019; Wang et al., 2020; Austin et al., 2021; Paperno et al., 2016; Rajpurkar et al., 2018).

For a full list of evaluation tasks that we used to evaluate StagFormer, see the Appendix (TODO).

## 4.2 RESULTS

We first present latency benchmarking results on accelerator hardware which demonstrate the gains we are able to see during decoding with StagFormer compared to a quality matched standard Transformers. The analysis is presented in Table 2.

At the 1-3 billion parameter scale, we compare shared-weights StagFormer to a baseline model with the same number of layers.

We also compare a model with double the number of Transformer layers with the separate-weights StagFormer which uses the same number of layers as the original baseline model in each pass. We chose to compare StagFormer to a Transformer with double the number of layers to compare the benefits of using staggered passes with adding more layers to the model.

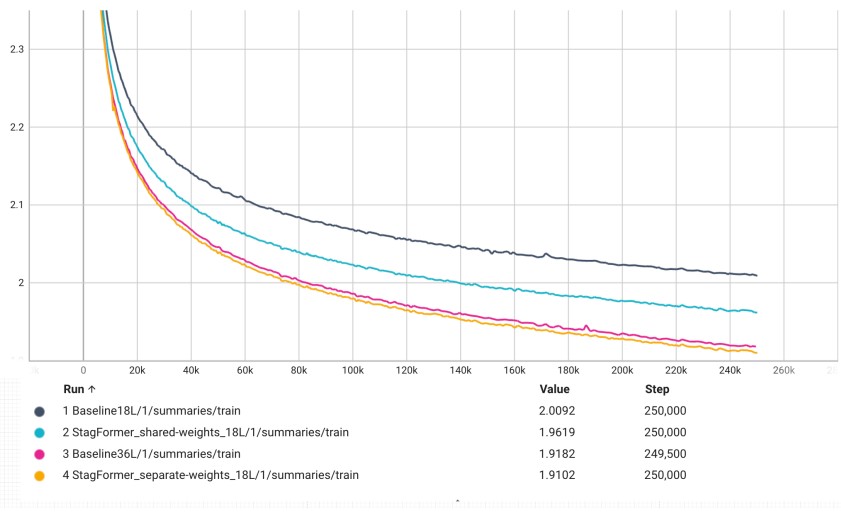

| Run ↑ | Value | Step |
|---|---|---|
| ● 1 Baseline18L/1/summaries/train | 2.0092 | 250,000 |
| ● 2 StagFormer_shared-weights_18L/1/summaries/train | 1.9619 | 250,000 |
| ● 3 Baseline36L/1/summaries/train | 1.9182 | 249,500 |
| ● 4 StagFormer_separate-weights_18L/1/summaries/train | 1.9102 | 250,000 |

Figure 4: Plot of the training loss for the 18 layer baseline (black), 18 layer shared-weights Stag-Former (blue), the 36 layer baseline (red), and separate-weights StagFormer with 2 stacks of 18 layers (yellow).

Table 3: Performance of Shared-Weight StagFormer pretraining and recurrent inference using Shared-Weight StagFormer

| Model | Pile Pplx. | HellaSwag | ARC-E | ARC-C | WinoGrande | SuperGLUE | MBPP | Lambada | SQuADv2 | GEM-XSum rouge2 | Avg. |
|---|---|---|---|---|---|---|---|---|---|---|---|
| Baseline (18L) 1.6B params | 4.026 | 49.8 | 60.1 | 31.8 | 53.4 | 59.3 | 0 | 3.7 | 31.8 | 0.9 | 32.3 |
| Baseline (36L) 2.8B params | 3.780 | 53.3 | 66.7 | 34.6 | 60.4 | 62.1 | 0.2 | 10.5 | 36.3 | 1.6 | 36.2 |
| StagFormer $p = 2$ Shared-Weights 18L Two-Networks 1.8B params | 3.896 | **54.3** | 61.7 | 31.7 | 57.7 | 59.5 | 0.2 | **10.4** | **46.9** | **2.1** | **36.1** |
| StagFormer $p = 2$ Shared-Weights 18L Recurrent 1.8B params | 3.896 | **54.3** | 61.7 | 31.7 | 57.7 | 59.5 | 0 | 4 | **42** | 0.4 | 34.6 |

### 4.3 RESULTS WITH LOCAL CROSS-ATTENTION

We also ran experiments using StagFormer with local cross-attention with both the separate- and shared-weights variants. We present results for experiments with local attention using window sizes 512, 128, and 1 in Table 5.

### 4.4 RESULTS WITH $p > 2$

We also present results from experiments with StagFormer with more than two stacks ($p > 2$). We show the effect of using more than two stacks on the shared-weights variant, and we show benchmarks for separate-weights StagFormer that use more than two passes to break the network layers into multiple passes. We also include ablations of using a linear combination of outputs for separate-weights StagFormer when $p > 2$ to demonstrate its impact on model quality. For shared-weights StagFormer, we match training during prefill and run all $p$ stacks, and then switch to recurrent inference for decoding. Note that for $p = 4$, some evaluation tasks failed due to memory constraints. We find that increasing $p$ surprisingly has a negative impact on model quality. See Table 3 for results.

Table 4: Performance of StagFormer on pretraining and a subset of evaluation tasks when $p > 2$

| Model | Train Pplx. | HellaSwag | ARC-E | ARC-C | WinoGrande | SuperGLUE | MBPP | Lambada | SQuADv2 | GEM-XSum rouge2 | Avg. |
|---|---|---|---|---|---|---|---|---|---|---|---|
| Baseline 18L 1.6B params | 4.026 | 49.8 | 60.1 | 31.8 | 53.4 | 59.3 | 0 | 3.7 | 31.8 | 0.9 | 32.3 |
| StagFormer $p = 3$ Shared-Weights 18L Recurrent 1.8B params | 3.858 | 51.3 | 55.6 | 31.8 | 59.6 | 59.1 | 0 | 3.8 | 21.5 | 1.1 | 31.5 |
| StagFormer $p = 4$ Shared-Weights 18L Recurrent 1.8B params | 3.870 | 46.6 | – | – | 51.9 | – | 0 | 0.2 | 5 | 0.6 | 17.4 |
| Baseline 2x Layers (36L) 2.8B params | 3.780 | 53.3 | 66.7 | 34.6 | 60.4 | 62.1 | 0.2 | 10.5 | 36.3 | 1.6 | 36.2 |
| StagFormer $p = 3$ Separate-Weights (3 x 12L) 3.0B params | 3.843 | 48.5 | 40.3 | 27.7 | 52.1 | 54.8 | 0.8 | 3.4 | 29.2 | 1 | 28.6 |
| StagFormer $p = 3$ Separate-Weights (3 x 12L) Sum-Outputs 3.0B params | **3.766** | 52.9 | 52.7 | 29.1 | 55.2 | 60 | 0 | 0 | 13.7 | 1 | 29.4 |
| StagFormer $p = 4$ Separate-Weights (4 x 9L) 3.0B params | 4.014 | 28.5 | 30.1 | 22.7 | 50.1 | 46.7 | 0 | 0 | 21.2 | 0 | 22.1 |
| StagFormer $p = 4$ Separate-Weights (4 x 9L) Sum-Outputs 3.0B params | **3.797** | 51.3 | 58 | 30.5 | 55 | 59.3 | 0 | 2 | 33.1 | 1.2 | **32.3** |

## 5 CONCLUSION

We present the StagFormer architecture as a way to increase the capacity of transformer models by allowing lower-level layers to attend to the final activations produced by the same or different networks. With separate-weights StagFormer, we demonstrate that we can use higher level representations of prior tokens to run data-independent transformer layers in parallel to process the current token without sacrificing quality.

### 5.1 FUTURE WORK AND LIMITATIONS

There are many aspects of the StagFormer architecture that are not well understood and requires future research. For example, training shared-weights StagFormer only approximates recurrent inference, since training requires a discrete number of passes. Furthermore, using shared-weights with more than 2 passes does not alleviate this issue. Future work could explore how to extend the

StagFormer algorithm that either better approximates or fully realizes recurrent decoding with better quality.

We also find that when we increase the number of stacks to more than two when using separate-weights StagFormer that the model's performance starts to degrade. Our experiment shows using a linear combination of the stacks' output helps the model recover a significant amount of quality, but not enough to match the fully sequential baseline with the equivalent number of layers. Later works could investigate whether it is possible to realize separate-weights StagFormer when $p > 2$ in order to further parallelize the execution of Transformer-based networks.

Another limitation is that cross-attention incurs additional quadratic computational cost in both time and space with respect to the input length. One way this work attempts to alleviate this additional cost is to use local cross-attention to stagger decoding between stacks. We show that it is possible to use the 512 window size, approximately fifty percent of the original context length, and suffer neg-ligible quality loss and even some improvements in downstream performance. However, we show that when the window size is decreased the performance of the StagFormerm model degrades. When the local cross-attention window is 1, cross-attention is linear with respect to input length instead of quadratic; however, the model quality suffers when the attention window size is restricted to such a small window. Other works can explore ways to reuse information-rich higher level activations in lower-level layers to allow parallel execution of layers in a way that incurs less computational cost than attention and matches a deeper model's quality.

One material limitation of StagFormer's parallel execution of layers is that it would require non-trivial communication cost to copy the result from one network over to the other. This prevents one from realizing the full theoretical latency benefit of running the StagFormer towers in parallel. Furthermore, since most models rely on the single program, multiple data (SPMD) paradigm (Xu et al., 2021), parallel execution of StagFormer stacks would require storing a copy of the token embeddings and final softmax tables in both cores when executing StagFormer stacks. Further work could explore how to extend this algorithm to help realize greater latency benefits when executing Transformer networks in parallel.

## 5.2 BROADER IMPACT

Transformer networks have mainly been used under the assumption that the execution of transformer layers must be done serially. StagFormer shows that it is possible to further parallelize execution of large language models by execution stacks of transformer layers in parallel and match the quality of a deeper model. StagFormer could help reduce the throughput latency of Transformer-based models, which allows these to be served at a lower cost. Efforts to lower the cost of deploying Transformer-based models has a large ecological and economic impact, since the amount of resources to deploy modern language models has become increasingly substantial.

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

# A  ADDITIONAL DETAILS ON STAGFORMER EXTENSIONS AND EXPERIMENTS

---

**Algorithm 3** Shared-weights StagFormer algorithm

---

**Input:** $\mathbf{t}_0^1, \ldots, \mathbf{t}_0^i \in \mathbb{R}^d$ : Token embeddings for positions $1, \ldots, i$ in the input sequence.
**Output:** $\mathbf{t}_l^i \in \mathbb{R}^d$ : The predicted token embedding for position $i + 1$ in the input sequence where $l$ is the total number of Transformer layers in the network.

1: **First pass** : for each layer $L_1, \ldots, L_l$ compute $\mathbf{t}_j^i = L_j \left( \mathbf{t}_{j-1}^{1,\ldots,i} \right)$.

Each application of $L_j$ using standard Transformer layer with self-attention and feed-forward layers.

2: **Second pass** : for each layer $L_1, \ldots, L_l$ compute $\mathbf{t}_j^i = L_j' \left( \mathbf{t}_{j-1}^{1,\ldots,i}, \mathbf{t}_L^{1,\ldots,i-1} \right)$.

Where $L_j'$ has an additional cross-attention layer between the self-attention and feed-forward layers to the Transformer layers in the first pass that uses $\mathbf{t}_l^{1,\ldots,i-1}$ for KV inputs.

3: **Return** $\mathbf{t}_l^i$.

---

---

**Algorithm 4** Separate-weights StagFormer $p > 2$ algorithm

---

**Input:** $\mathbf{t}_0^1, \ldots, \mathbf{t}_0^i \in \mathbb{R}^d$ : Token embeddings for positions $1, \ldots, i$ in the input sequence.
**Output:** $\mathbf{t}_\ell^i \in \mathbb{R}^d$ : The predicted token embedding for position $i + 1$ in the input sequence where $\ell$ is the total number of Transformer layers in the network.

1: **First pass** : for each layer $L_1, \ldots, L_h$ where $h \equiv \lfloor \ell/p \rfloor$ compute $\mathbf{t}_j^i = L_j \left( \mathbf{t}_{j-1}^{1,\ldots,i} \right)$.

Each application of $L_j$ using standard Transformer layer with self-attention and feed-forward layers.

2: **Subsequent passes** : for each $k \in \{2, \ldots, p\}$ do:

for each layer in $L_{h \cdot (k-1)+1}, \ldots, L_{h \cdot k}$ compute $\mathbf{t}_j^i = L_j' \left( \mathbf{t}_u^{1,\ldots,i}, \mathbf{t}_{h \cdot (k-1)}^{1,\ldots,i-1} \right)$.

Where $u = 0$ when $j = h \cdot (k-1) + 1$ and $u = j$ otherwise.

Where $L_j'$ is a Transformer layer that has an additional cross-attention layer between the self-attention and feed-forward layers that uses $\mathbf{t}_{h \cdot (k-1)}^{1,\ldots,i-1}$ for KV inputs.

3: **Return** $\sum_{k}^{p} \alpha_k \cdot \mathbf{t}_{h \cdot k}^i$.

Where each $\alpha_k$ is a learnable scalar.

---

Table 5: Performance of StagFormer on pretraining and eval tasks with local cross-attention

| Model | Pile Pplx. | HellaSwag | ARC-E | ARC-C | WinoGrande | SuperGLUE | MBPP | Lambada | SQuADv2 | GEM-XSum rouge2 | Avg. |
|---|---|---|---|---|---|---|---|---|---|---|---|
| Baseline 18L 1.6B params | 4.026 | 49.8 | 60.1 | 31.8 | 53.4 | 59.3 | 0 | 3.7 | 31.8 | 0.9 | 32.3 |
| StagFormer Shared-Weights Window 512 Two-Networks 1.8B params | 3.908 | **55.7** | 64.9 | 33.9 | 59.4 | 60.1 | 0 | **22** | **39.4** | 1.6 | **37.4** |
| StagFormer Shared-Weights Window 512 Recurrent 1.8B params | 3.908 | **55.7** | 64.9 | 33.9 | 59.4 | 60.1 | 0 | 9.3 | 38 | 1.1 | 35.8 |
| StagFormer Shared-Weights Window 128 Two-Networks 1.8B params | 3.929 | **56.4** | 64.9 | 34 | 59.4 | 59.8 | 0.2 | **31.3** | **40.3** | **1.8** | **38.7** |
| StagFormer Shared-Weights Window 128 Recurrent 1.8B params | 3.929 | **55.7** | 65.3 | 34.5 | 59.5 | 61 | 0 | 8.1 | **42.5** | **2.1** | **37.5** |
| StagFormer Shared-Weights Window 1 Two-Networks 1.8B params | 3.951 | 46.8 | 56.5 | 29.4 | 58.5 | 58 | 0 | 0.2 | 34.8 | 0.6 | 31.6 |
| StagFormer Shared-Weights Window 1 Recurrent 1.8B params | 3.951 | 46.8 | 56.5 | 29.4 | 58.5 | 58 | 0 | 0.2 | 34.8 | 0.6 | 31.6 |
| Baseline 2x Layers (36L) 2.8B params | 3.780 | 53.3 | 66.7 | 34.6 | 60.4 | 62.1 | 0.2 | 10.5 | 36.3 | 1.6 | 36.2 |
| StagFormer Separate-Weights Window 512 2.9B params | 3.767 | **58.6** | **68.2** | **36.9** | **61.8** | **63.3** | 5 | **33.6** | **41.5** | **1.9** | **41.2** |
| StagFormer Separate-Weights Window 128 2.9B params | 3.797 | 51.3 | 55.6 | 32.8 | 59.6 | 59.1 | 0 | 3.8 | 21.5 | 1.1 | 31.6 |
| StagFormer Separate-Weights Window 1 2.9B params | 3.818 | 33.3 | 30.9 | 25.3 | 51.2 | 45.6 | 0 | 0 | 0 | 0 | 20.7 |

