# OpenReview forum: "StagFormer:  A Staggered Transformer for Decoding Layers in Parallel"
_ICLR.cc/2025/Conference — Submitted to ICLR 2025_

### Official Review · Reviewer_Apqd · 2024-11-03

**Soundness:** 3
**Presentation:** 2
**Contribution:** 3
**Rating:** 5
**Confidence:** 4

**Summary:**

This paper proposes a novel transformer architecture that effectively reduces the number of sequential steps (layers) during the decoding process by staggering the computation across different time-steps. This allows for improved parallelism during decoding individual sequences, providing speedups during inference.

**Strengths:**

- staggered computation leads to significant improvements in per-time-step decoding speeds while slightly improving performance
- provides results and analysis of different variants of staggered transformers that further explores the architecture's efficacy

**Weaknesses:**

- Biggest critique is that it lacks comparative analysis of staggering computation vs. simply increasing the width of the model and lowering the number of layers, as this increases per layer parallelism while decreasing the number of layers leading to a similar improvement in decoding speed.
- This technique is possibly only useful for speeding up decoding when only a single sequence is being decoded. A non-staggered model could in theory process twice the batch size as it has half the parallelism (and hence half the per layer memory requirement) of a model staggered with p=2.
- StagFormer is possibly slower to train (as inferred by its slower pre-filling speed)
- Paper could be further refined (minor critique):
    - Some references are configured incorrectly (Table ?? in page 5, "TODO" in page 8)
    - Plots have unnecessary information (Figure 4 doesn't need texts like /1/summarize/train)

**Questions:**

Addressing the weaknesses outlined above would improve the paper.

---

> ### Author Response · Authors · 2024-11-28
> **Response to Reviewer Apqd**
>
> - **Comparative analysis vs simply increasing width of the model**: This is an interesting question. However, there is empirical evidence building every day that depth is crucial for reasoning oriented downstream tasks. Although a wider shallower model might achieve a similar log pplx during training it tends to be less performant on downstream tasks compared to deeper models.
> - **Technique only useful when a single sequence is being decoded**: We would like to emphasize that this is false. Our architecture naturally extends to handle processing for a batch of sequences at once.
> - **Slower to train**: Indeed, this is the case. However, in many applications today, it is perfectly acceptable to have a model that is slower to train but more performant during inference since the training cost is a one-time cost. For instance, take the popular Llama 3 series of models wherein the smallest models were trained for much longer than is suggested by data-scaling laws with the intention that they will be used repeatedly for inference so a one-time higher training cost is easily justified.
> - **Writing typos/ polishing of the paper**: We apologize for the typos and missing references. We will fix all of them and also polish the presentation in the paper to make it more detailed and include better explanations and insights from the results presented in our work.

---

> > ### Comment · Reviewer_Apqd · 2024-11-28
> >
> > Thank you for the response, I would like the authors to clarify a few follow up points:
> >
> > > there is empirical evidence building every day that depth is crucial for reasoning oriented downstream tasks. Although a wider shallower model might achieve a similar log pplx during training it tends to be less performant on downstream tasks compared to deeper models.
> >
> > If there is clear evidence from prior works, can the authors point to specific studies that substantiates these claims and include them in the paper?
> >
> > > We would like to emphasize that this is false. Our architecture naturally extends to handle processing for a batch of sequences at once.
> >
> > The point of the original question wasn't about whether the model can handle a larger batch size. But rather, given limited resources, a model that has greater degrees of parallelism (stagformer) will require more memory. Therefore, a non-staggered model can potentially process twice as many sequences in parallel as a staggered model with P=2 (twice the parallelism). For this reason, stagformer seems to only improve generation speed when there is an expected fixed batch size. I recommend the authors to clarify this, and simultaneously mention that for edge applications, batch size is typically fixed to 1. I believe this would improve the way the paper is presented.

---

### Official Review · Reviewer_TMDV · 2024-11-03

**Soundness:** 2
**Presentation:** 2
**Contribution:** 2
**Rating:** 3
**Confidence:** 4

**Summary:**

This paper proposes a novel Transformer architecture called StagFormer designed to improve the efficiency of decoding in Transformer-based language models by enabling the parallel execution of layers along the depth axis.

**Strengths:**

1. StagFormer introduces a unique method to break the sequential dependency of layers in Transformers, enabling parallel execution.
2. Experiments demonstrate significant latency reduction while maintaining or even exceeding the quality of a standard Transformer.
3. The paper investigates different StagFormer variants, offering flexibility and adaptability to various scenarios and resource constraints.
4. The paper effectively explains the StagFormer concept and its variants, supported by clear diagrams and algorithms.

**Weaknesses:**

1. Limited exploration of p > 2. While the paper explores StagFormer with more than two stacks, it acknowledges performance degradation and the need for further research in this area.
2. The paper mentions the communication cost associated with parallel execution but doesn't offer concrete solutions to mitigate it.
3. While the Pile dataset is comprehensive, evaluating on additional datasets would strengthen the generalizability of the findings.
4. Comparing StagFormer with other methods for efficient Transformer inference, such as speculative decoding, would provide a more comprehensive perspective.

**Questions:**

1. How does varying the depth of individual stacks in StagFormer affect the trade-off between decoding speed and model quality?
2. What factors determine the optimal number of stacks for a given application, balancing computational efficiency and performance?
3. Could the staggering concept be extended to encoder-decoder Transformers, like those used in machine translation?
4. How well could StagFormer be combined with other techniques, like quantization or knowledge distillation, to further enhance decoding efficiency?

---

> ### Author Response · Authors · 2024-11-28
> **Response to Reviewer TMDV**
>
> - **Limited exploration of p>2 setting**: Since this is the first paper introducing this idea, we wanted to focus more on simpler settings where we found the idea to give strong benefits. We tried extending the idea naturally to p>2 and found diminishing returns. Indeed, this is to be expected as one can imagine that for a very large p, the staggering basically forces a small Transformer network to predict p tokens into the future which becomes information theoretically impossible as p increases. (For instance, it might involve answering a question before the question is even asked). We do expect the performance benefits to drop off after some small value of p. We believe the exploration of techniques to avoid this dropoff is out of scope of the current paper.
> - **Mitigating communication cost associated with parallel execution**: This is an important issue and is a natural by-product of the using the StagFormer idea to speed up execution. Mitigating this can be achieved by optimized hardware setups which enable fast inter-chip communication. This is outside the scope of the current paper though.
> - **Additional datasets apart from Pile**: We thank the reviewer for raising this point. For language modeling, we believe the Pile is a very comprehensive dataset. In fact, some of the well-known contemporary research papers such as “Mamba: Linear Time Sequence Modeling with Selective State Spaces” rely on pre-training experiments on the Pile. Pre-training experiments on a dataset the size of Pile take a significant amount of compute resources already and it is not easy to scale up to larger datasets on a limited compute.
> - **Comparison with other methods for efficient Transformer inference**: Indeed there is plenty of research on making Transformer inference more efficient including the methods the reviewer pointed out such as speculative decoding, quantization, knowledge distillation among others. We don’t view StagFormer as competing with these other methods, rather StagFormer can be used in conjunction with most of these methods (very naturally with techniques such as knowledge distillation and quantization) to give stronger gains. Due to the tremendous amount of research on methods for optimizing Transformer inference, it is hard to perform a comparative analysis with each of them. We will try to include comparative experiments with a few well-known methods in future drafts.
> - **Effect of depth on decoding speed and model quality**: Increasing the depth of each StagFormer stack would make the model of higher quality but it would decode slower.
> - **Factors influencing the optimal number of stacks**: This is a good question. Given a quality threshold that we want to achieve, we believe a small number of stacks (many times just 2) is optimal as too many stacks can start causing severe quality degradations.
> - **Extending the staggering idea to encoder-decoder transformers**: The staggering idea can be used in the decoder of an encoder-decoder architecture naturally.

---

### Official Review · Reviewer_oS5P · 2024-11-03

**Soundness:** 2
**Presentation:** 3
**Contribution:** 3
**Rating:** 3
**Confidence:** 4

**Summary:**

The authors present the Staggered Transformer (StagFormer) and its variants which relieve sequential dependancies in the decoding pipeline to enable higher levels of parallel execution.

Consider a transformer with two stacks of layers, A (bottom half) and B (upper half). In vanilla transformers, the input token embedding is passed to stack A. Then, the output of stack A is passed to stack B. All layers apply self-attention on outputs of the previous layer.

In the baseline StagFormer (`Separate-Weights`), stack A is the same. However, stack B takes in the input token embedding rather than the output of stack A.
To supplement this, stack B applies cross-attention on the final outputs of stack A, up until the previous token. In other words, stack B cross-attends to the outputs of *all previous input tokens* from stack A, instead of directly inputting that of the *current* input token. This relieves the dependency of stack B on stack A, within a single decoding step, thus both A and B can be computed simultaneously.

The authors investigate many variants of this design:
1. `Shared-Weights`: this is where stack A and stack B share the same model parameters (excluding the cross-attention layers which are unique to stack B).
2. `Recurrent, Shared-Weights`: this is a unique decoding method for the `Shared-Weights` trained model. In `Shared-Weights` stack A and B are identical, except that stack B applies cross-attention to outputs from stack A. Essentially, the shared stack S (= A = B) is first forwarded without cross-attention, and then forwarded a second time *with* cross-attention, attending to outputs from the first forward pass. The `Recurrent` setting refers to that where the first forward pass is skipped, and cross-attention in the second pass attends to outputs of the "second" pass from the previous decoding step.
3. `p > 2`: this is where more than two stacks are considered.

When compared to vanilla transformers pretrained from scratch, StagFormers show various advantages, mainly:
- `Shared-Weights 2x18L`: StagFormer outperforms the vanilla 18L baseline (with roughly same parameters) in both perplexity and average task performance. Using recurrent decoding (roughly matching 18L baseline computation), average task performance lies between the two. StagFormer underperforms the vanilla 36L baseline with roughly same computation in perplexity, but performs comparably on tasks.
- `Separate-Weights 2x18L`: StagFormer outperforms the vanilla 36L baseline (with roughly same parameters and compute) in both perplexity and task performance.

**Strengths:**

1. The idea and architecture design are very novel
1. The authors propose numerous variants which showcase the potential extension of the idea across various axes–parallel execution, weight sharing, recurrent computation.
1. The architecture shows clear advantages over vanilla transformers across its variants
1. The writing is easy to follow and visual depiction of the architecture and its variants are superb.

**Weaknesses:**

1. **Memory  bottlenecks during decoding may hinder benefits of parallel execution, which is not discussed**: LM decoding is typically bottlenecked by memory rather than compute (see references below). When batch size x context length is small, memory access is dominated by model parameter access. Otherwise, memory access is dominated by KV cache access. While StagFormer can conceptually *parallelize* execution of layers, the associated memory access load cannot be parallelized. In fact, the cross-attention layer will add additional KV cache access overhead. These are critical to assessing the actual wallclock time benefits of decoding with StagFormers compared to vanilla transformers, but is not discussed.
    1. Different variants of StagFormers will have different memory bottlenecks. Examples:
        1. All variants: cross-attention is added in half of layers. Therefore, the overall KV cache access overhead will increase by 50% (relative to that of self-attention, used in all layers). This will have a larger effect on decoding time as batch size x sequence length becomes large.
        1. `Separate-Weights`: both stacks can be executed in parallel, but the memory load is identical as the parameters of both stacks must be retrieved from memory. This means that wall-clock time should typically be identical to vanilla transformers, as decoding is bottlenecked by memory access. `Shared-Weights` can solve this issue.
    1. **It is unclear which StagFormer variant is used in Table 2, raising questions on the performance vs latency comparison**: While Table 2 states that a "comparable quality StagFormer" is 33% faster than baseline transformer during decoding, the exact variant is unclear. Given the reasons above, it seems likely that this is the `Shared-Weights 2x18L` variant. While its average task performance is comparable to baseline 36L, its PPL is in the middle of that between vanilla 18L and 36L. It would be misleading to describe this variant as "comparable quality" to vanilla 36L.
    1. **Missing comparison of performance vs latency across model variants**: Expanding on the point above, a comparison of prefill/decode time across model variants will provide a clear picture on the performance vs latency benefits of each model variant. This could take the form of a single table that lists the PPL, task performance, and prefill/decode time for each model. In the case of  `p > 2, Shared-Weight` variants, I believe this may actually reveal some advantages in terms of latency.
    1. **The additional KV cache overhead of cross attention may slow down decoding for longer contexts**: Since KV cache overhead is quadratic to context length, the decode time advantages as shown in Table 2 will likely diminish with longer contexts, especially in batch decoding. Given the relatively short context length of 1024 tokens considered in this study, compared to modern LLMs with 8K+ context, measurement on longer contexts and larger batch sizes can help gauge the potential of the architecture.
1. **Misleading task performance of `Recurrent` variant**: In Table 3 (for example), the performance of various tasks are identical between the `Shared-Weights 18L` model and its `Recurrent` counterpart. This is likely because the tasks are measured in a teacher-forcing setting, where the outputs of the prefill stage are used for evaluation. This does not represent the task performance of the `Recurrent` setting, as recurrence is only applied to decoding, as explained in Section 3.2.
1. **Experimental results on model variants are hard to follow**: The organization of the results section could be improved to make the comparison between different model variants more clear.
    1. Within tables, variations could be better indicated with separate columns, task names could be shortened for space, latency metrics could be included, etc.
    1. Results on different variants are presented in multiple tables without a clear organization.
1. **Incomplete writing**: "(TODO)" in Line 385, the reference error "??" in Line 267, and numerous typos suggest that this is an incomplete manuscript that is not ready for review.

References on memory bottlenecks during inference
- [Efficiently Scaling Transformer Inference](https://arxiv.org/abs/2211.05102)
- [LLM Inference Unveiled: Survey and Roofline Model Insights](https://arxiv.org/abs/2402.16363v4)
- [Taming Throughput-Latency Tradeoff in LLM Inference with Sarathi-Serve](https://arxiv.org/abs/2403.02310)
- [Block Transformer: Global-to-Local Language Modeling for Fast Inference](https://arxiv.org/abs/2406.02657)

**Questions:**

1. Can you describe the architecture shape (vocab size, qkv heads, embedding dimensions) and its justification? The vocab size of 256K is quite high for models of this size.
1. In Lines ~499-501, the authors mention that cross-attention is linear to input length instead of quadratic with window size 1. Isn't it linear with any fixed window size? Considering that the cost of attention mainly stems from KV cache IO during decoding, I think the constant factor with a window size as small as 128 makes the cost of cross-attention negligible compared to self-attention (especially when expanding to modern context lengths of 8K or more).
    1. However, the *increase* in performance when going from full cross-attention (1024) to windowed attention with window size 512 and 128 is strange. Can the authors justify this increase in performance?

---

> ### Author Response · Authors · 2024-11-28
> **Response to Reviewer oS5P**
>
> We thank you for your review and feedback on our paper. We hope to address your concerns below.
>
> - **Memory bottlenecks during decoding**: Thank you for these questions. We will add a detailed discussion of this topic to clarify some possible misconceptions with our current write-up. We agree that memory access can dominate decoding times. Irrespective of whether we are in a small batch size/context length setting or a large batch size/context length setting, instead of operating a size B Transformer model in parallel on C accelerator chips, we propose serving a size B (approximately) StagFormer model on 2C chips which will reduce the latency of decoding each token without adding additional memory load `on any given chip`. The use of extra hardware chips allows us to be memory usage neutral with respect to the baseline model. Our simulated latency measurement takes this memory load into account. In particular, your claim that “While StagFormer can conceptually parallelize execution of layers, the associated memory access load cannot be parallelized.” doesn’t hold here since the additional hardware chips help with the extra memory access required for the cross attention KV caches.
> - **Stagformer variant used in table 2**: We apologize for the lack of clarity here. The variant we used here is in fact the Separate Weights - **Stagformer 2x18L model** We don’t run into the memory issues you mentioned because of the explanation given above. Note that we do take a hit compared to the theoretical latency savings of 50% (and only achieve ~33%) because of the additional cross-attention and a few other engineering overheads.
> - **Misleading task performance of the Recurrent variant**: We apologize for not making this more clear. We will explicitly mention that on scoring tasks recurrent decoding doesn’t make a difference and the performance remains the same since prefill still goes through 2 stacks.
> - **Incomplete references**: We apologize for this oversight. We will fix all typos, missing references and also make a pass over the entire paper to polish the writing and clarity of the results.
> - **Experimental results on model variants are hard to follow**: We apologize for the confusion and will re-organize the sections to make the reading more clear. We will also break down the results into more subsections so as to highlight the main insights within each table more directly.
> - **Architecture details**: A vocab size of 256k although uncommon for models of this size, is not completely unheard of. For instance, Gemma which has models of a similar size used 256k vocab as well. Our 18 layer baseline uses 1536 model dimension, 12288 as hidden dimension and 12 attention heads. We will add these details to the paper.
> - **Confusion on time complexity of computing cross-attention**: We apologize for this confusion. We were referring to the time complexity of attention computation during the prefill step wherein we need to compute contextual embeddings for every token in the sequence. This is quadratic normally and can be made linear when choosing a small window size. Consequently, during decoding, the linear cost of attention during decoding steps drops to constant with a constant window size. We will clarify this in the writeup.

---

> > ### Comment · Reviewer_oS5P · 2024-11-28
> >
> > Thank you for addressing my concerns.
> >
> > - **Memory bottlenecks during decoding**: Thank you for the clarification. However, the paper itself does not mention parallelization across multiple chips, except for the fact that 16 TPU chips were used for measurements. If this parallelization is the main advantage of StagFormers, it should be discussed in relation to previous methods. Relevant topics include: challenges with multi-device parallelization of transformers; existing techniques such as tensor parallelism; detailed explanation of how model parameters and KV cache of StagFormers are parallelized across devices and how this outperforms existing methods in mitigating key costs (compute, memory, and/or communication)–at least in principle; comparison with vanilla transformers using existing parallelism strategies. In the current experiments, it is not clear what parallelism strategies were used for each model.
> >   - `instead of operating a size B Transformer model in parallel on C accelerator chips, we propose serving a size B (approximately) StagFormer model on 2C chips`: how much better/faster is StagFormer compared to using 2C chips with existing parallelism techniques on vanilla transformers?
> > - **Misleading task performance of the Recurrent variant**: I think it is misleading to attribute these scores to the recurrent decoding variant, as it does not represent the performance of the model in generation (while models of this size and training length are typically not used for generation, I think the expectation is that these scores are used to gauge the potential of the architecture when they are eventually scaled up). Only evaluations that use decode-stage outputs should be attributed to the recurrent decoding variant.
> > - **Architecture details**: Thank you for the clarification.

---

### Official Review · Reviewer_vLV7 · 2024-11-05

**Soundness:** 2
**Presentation:** 1
**Contribution:** 3
**Rating:** 3
**Confidence:** 4

**Summary:**

This paper proposed a new architecture StagFormer, which stagger the time dependency between the lower and upper layers. The overall design seems a little non-intuitive, but has a lot of potential for throughput and performance. For example, parameter sharing or local cross-attention could yield better throughput.

**Strengths:**

- StagFormer architecture is interesting, and has very good potential for both performance and throughput.
- The idea of parameter sharing and recurrent decoding looks good.

**Weaknesses:**

- I like the concept and potential of this paper, but I believe that this paper is not well-organized, and looks like unfinished work yet. For example, there is missing reference in L.267 (I guess this refers to Table 3), there are a few results for proof of concept.
- Table 3 is showing few-shot results for gray, blue, red lines in Figure 4 (correct me if I’m wrong.) I wonder why shared-weights StagFormer (blue) outperforms Baseline 2.8B params (red) in some benchmarks, even though it shows higher loss values.
- What makes StagFormer 2.9B to outperform Baseline 2.8B params in Table 1? Is it due to cross-attention in upper layers? This looks somewhat interesting and also confusing because I thought the changed structure (using previous timestep’s intermediate activations) could degrade performance a lot.
- How did the authors measure the decoding time in Table 2? Running separate parameters in parallel is not trivial, I believe. Is it actual time or hypothetical time by assuming parallel execution of them?

**Questions:**

- For KV-caches, the total KV caches are a little increased by the amount of one layer for cross-attention in upper layers, rights?

---

> ### Author Response · Authors · 2024-11-27
> **Response to reviewer vLV7**
>
> We would like to thank the reviewer for their careful reading of our paper and for the feedback provided. We try to address the concerns raised below.
>
> - **Typos/Paper organization and polishing**: We apologize for the typos and missing references. We will fix them. We will also take a pass to make the presentation of our main results cleaner and more detailed and polished. Can you elaborate on what you mean by “few results for proof of concept”?
> - **Table 3: Shared Weights Stagformer (1.6B) outperforming 2.8B Baseline on downstream evaluations**: While it appears that the shared weights stagformer outperforms the 2.8B baseline model on some evaluations, it is not a universal trend and not hence inconclusive whether it is better than a 2.8B model. Moreover, in general we don’t expect it to outperform a 2x depth baseline. That is why we didn’t focus too much on the specific numbers on a few evals where the stagfomer model was outperforming the 2x depth baseline.
> - **Table 1: Strong performance of Stagformer 2.9B model in comparison to Baseline 2.8B model**: As you correctly point out, the strong performance of the StagFormer 2.9B model in comparison to the Baseline 2.8B is due to the presence of the cross attention layers. In general, we expect it to roughly match the performance of the baseline, i.e. the extra cross-attention layers help offset the quality loss from staggering. Indeed, this can be seen in Table 1 where the improvements we see in some columns are very minor over the 2.8B Baseline.
> - **Measurement of the decoding time**: We apologize for the lack of details in this section. We will add more discussion in this regard. We measured the decoding time using a setup that simulates a Stagformer with 2 stacks running on twice the number of chips used by a baseline model. While we faithfully account for every segment of the StagFormer model, we ignore the inter-chip communication cost between the first and second stacks of the Stagformer. This communication cost is expected to be minimal in practice compared to the other contributor to latency under an optimized hardware setup.
> - **KV-cache question**: Yes for half of the layers, we would need an extra KV-cache for the cross-attention. Note however that in the setup described above where we use double the number of chips as a baseline model this would not increase the per chip memory load.

---

> ### Comment · Reviewer_vLV7 · 2024-11-29
>
> Thanks for clarifying some misleading points.
>
> **1.** I clarified my words: "The experiments provided seem limited in scope, hinting at strong performance only in a few simplified settings. To further validate the robustness, a more extensive evaluation and delicate ablation studies are recommend like exploring across various model sizes, architectures.".
>
> **2.** Thanks.
>
> **3.** I believe that StagFormer has a great potential by looking at the overall results.
>
> **4.** Thanks.
>
> **5.** Thanks.
>
> While I truly believe in the potential of this work and its contribution, I maintain my score as I feel it's not yet ready for publication.

---

### Meta-Review · Area_Chair_94ro · 2024-12-22

**Metareview:**

The authors proposed an interesting idea that can possibly result in throughput gains via staggered computation and parameter sharing. However, the manuscript is far from being polished, with missing references, ambiguous variant comparisons, and inadequate consideration of memory overhead that compromises parallel decoding gains. The unclear performance vs. latency tradeoffs, along with limited exploration of longer contexts and direct comparisons to other efficient approaches, further weaken the conclusions. Overall, I think the idea seems interesting, but a much more thorough study is needed. Also, the downsides of the proposed approach (reduced depth and how it affects reasoning, etc.) needs to be carefully studied.

**Additional Comments On Reviewer Discussion:**

While the authors addressed some minor concerns, the reviewers' major concerns remained after the rebuttal.

---

### Decision · Program_Chairs · 2025-01-22

Reject